# Emerging Methods for Integrative Management of Chronic Diseases: Utilizing mHealth Apps for Lifestyle Interventions

**DOI:** 10.3390/nu17091506

**Published:** 2025-04-29

**Authors:** Alina Spinean, Alexandra Mladin, Simona Carniciu, Ana Maria Alexandra Stănescu, Cristian Serafinceanu

**Affiliations:** Department of Diabetes, Nutrition and Metabolic Diseases, Carol Davila University of Medicine and Pharmacy, 050474 Bucharest, Romania; alinaspinean@yahoo.com (A.S.);

**Keywords:** mHealth, technology, health apps, nutrition, lifestyle, obesity, diabetes

## Abstract

**Background/Objectives:** Mobile health (mHealth) apps have become a revolutionary tool in managing and treating chronic diseases, providing numerous advantages for both patients and healthcare providers. These apps leverage technology to offer a variety of functions that support the monitoring, management, and enhancement of a patient’s health. **Methods:** We performed an observational study with 147 participants, using a questionnaire to evaluate the impact of mHealth applications on lifestyle changes in individuals managing chronic health conditions, including diabetes, obesity, and hypertension. **Results:** The study found that 40% of participants used the app daily, with a further 24.39% using it weekly and 14.63% using it occasionally. The positive health impact of the app was evident, with improvements in key health metrics such as glucose levels (73.42%), weight (62.02%), and adherence to dietary recommendations (71.31%). **Conslussions:** These findings aligned with studies on the effectiveness of mHealth apps in managing chronic conditions like diabetes. These broad health improvements reported by users suggested that the app was effective in promoting healthier behaviors. The high levels of user satisfaction and engagement highlighted how effective the app was. All in all, our study found that mHealth apps are valuable tools for people managing chronic health conditions, helping to motivate users and improve their health.

## 1. Introduction

Mobile health (mHealth v1.5.3) apps have emerged as a significant tool in managing chronic diseases, offering valuable benefits for both patients and healthcare providers [1]. As chronic conditions such as diabetes and obesity become increasingly prevalent, these apps have proven essential in helping individuals implement lifestyle changes that lead to improved health outcomes. By leveraging smartphones and wearable devices, mHealth apps facilitate real-time monitoring, provide personalized interventions, and offer actionable feedback, empowering patients to manage their conditions more effectively [2].

Lifestyle modifications are crucial for managing both diabetes and obesity. For diabetes patients, maintaining stable blood glucose levels through a balanced diet, regular exercise, and medication adherence is essential to minimize complications and enhance quality of life [3]. In the case of obesity, successful weight management often requires sustained behavioral changes, such as better eating habits and increased physical activity. However, achieving these changes can be challenging for many patients, as it requires ongoing effort, motivation, and support [4].

### 1.1. mHealth Apps and Their Functionality

mHealth apps, or mobile health applications, are software tools designed to help manage health conditions through mobile devices such as smartphones, tablets, and wearables. These apps utilize technology to provide various functions that support the monitoring, management, and improvement of a patient’s health. While the specific features of mHealth apps may vary, they commonly include tracking vital signs, managing medication schedules, offering lifestyle recommendations, and enabling communication between patients and healthcare providers [1,2]. For individuals managing chronic conditions like diabetes and obesity, these apps play a crucial role in encouraging lifestyle changes and helping patients stay engaged with their treatment plans.

### 1.2. Types of mHealth Apps


Tracking Apps: These apps assist patients in monitoring key health metrics like blood glucose levels, physical activity, and caloric intake. For example, diabetes patients often use glucose monitoring apps to log their blood sugar levels throughout the day, helping them identify trends and triggers [5]. Obese patients may use weight-tracking apps to track their progress toward weight loss goals while receiving feedback on their diet and exercise habits [6].Educational Apps: These apps offer patients valuable information on disease management, lifestyle changes, and self-care. They educate users about healthy diet choices, exercise routines, and medication adherence. Many mHealth apps provide personalized content based on an individual’s specific health condition and goals [7]. For instance, diabetes management apps often feature educational tools on carbohydrate counting and strategies to manage blood glucose fluctuations.Medication Reminders: A key feature for diabetes and obesity patients is medication adherence. Many mHealth apps include reminder systems that notify users to take their medications at specific times. This is especially important for diabetes patients who need to manage insulin doses and blood glucose-lowering medications consistently [5].Patient–Provider Communication: Some mHealth apps enable secure messaging between patients and healthcare providers, allowing users to discuss symptoms, adjust treatment plans, and receive professional guidance without the need for in-person visits. This feature is particularly valuable for individuals managing chronic conditions who require continuous monitoring and support but face challenges accessing healthcare services regularly [8].


### 1.3. Technological Features

mHealth apps incorporate several technological features to enhance their effectiveness in managing chronic conditions:**Real-time Monitoring**: By integrating wearable devices like glucose meters or fitness trackers, mHealth apps enable continuous monitoring of vital health metrics. This allows patients to keep track of their condition at all times and take action when necessary [4].**Data Integration and Analysis**: Many apps collect data from various sources, such as wearables and user input, to generate reports or graphs. This integration helps users easily track their health progress and identify behavioral trends that may influence their health outcomes [5].**Personalization**: Using algorithms, mHealth apps often tailor content and advice to individual users. For example, a weight loss app might adjust diet and exercise plans based on a user’s progress and preferences [7]. This personalization boosts patient engagement and improves adherence to health recommendations.**Gamification and Social Features**: To enhance patient engagement, many apps include gamification elements, such as rewards, badges, and challenges. Some also have social features, where users can share their progress or join community challenges, creating a sense of accountability and motivation [9].

In addition, mHealth apps use artificial intelligence (AI) and machine learning to adapt to users’ habits, offering more tailored and refined health insights. This is particularly beneficial for managing chronic conditions like diabetes and obesity, where ongoing adjustments to lifestyle factors are crucial.

### 1.4. Clinical Evidence Supporting mHealth Functionality

Several studies have shown that mHealth apps can have a positive impact on lifestyle changes and help manage chronic conditions. For instance, research by Free et al. in 2013 highlighted how mHealth apps improved medication adherence in patients with chronic diseases like diabetes, with users being more likely to stick to their prescribed treatment plans [1]. Similarly, a systematic review conducted by Wang et al. in 2020 found that mHealth interventions were effective in encouraging weight loss and increasing physical activity in patients with obesity [10].

### 1.5. Challenges and Limitations of mHealth Apps

Despite the potential benefits, mHealth apps faced several challenges in their implementation. One major issue was user engagement. While many patients initially adopted the apps, a significant number discontinued use after a short period. This was primarily due to lack of motivation or difficulties in adapting to the technology. Furthermore, data privacy and security concerns emerged, as mHealth apps often collected sensitive health data. Not all apps were compliant with regulatory standards for data protection, which led to hesitation among users regarding data sharing [11].

Another challenge identified was the variability in app quality. Many apps lacked clinical validation, raising doubts about their efficacy in managing chronic conditions such as diabetes and obesity. To address these limitations, it is essential for healthcare providers and app developers to collaborate, ensuring the development of more reliable, evidence-based applications. Robust user support systems would also be crucial in enhancing user engagement and satisfaction [12].

This study explored the impact of mHealth apps on lifestyle changes among patients with diabetes and obesity. It specifically examined how these apps influenced key behaviors such as dietary choices, physical activity, and medication adherence. The findings also highlighted how mHealth apps contributed to the overall management of these chronic conditions. Through an observational approach, this paper aimed to underline both the benefits and challenges of using mHealth apps for supporting lifestyle modifications in individuals managing diabetes and obesity [11].

### 1.6. Purpose of the Study

The purpose of this study was to explore the impact of mHealth applications on lifestyle changes in individuals managing chronic health conditions such as diabetes, obesity, and hypertension. The study specifically aimed to assess the frequency of use, user engagement, and the perceived usefulness of these apps in influencing lifestyle changes like diet, physical activity, and medication adherence. Additionally, the study sought to identify the most appreciated features of mHealth apps, examine user satisfaction levels, and highlight the challenges users faced.

### 1.7. Objectives


To assess the demographic profile of users engaging with mHealth apps for chronic health management—Investigated the demographic characteristics (age, gender, and education) of participants using the app.To evaluate the perceived effectiveness of mHealth apps in promoting positive health outcomes—Assessed users’ self-reported health improvements (e.g., glucose levels, weight, and diet adherence) and the perceived impact of the app on their ability to manage their health conditions.To identify the most valued features of mHealth apps according to users—Investigated user preferences regarding specific app features such as reminders, tracking tools, and design, and how these features contributed to their overall experience and health management.To assess user satisfaction with the app’s design and functionality—Measured user satisfaction, with the app’s design, usability, and performance, and identified areas for improvement, including concerns such as excessive advertisements and the desire for additional features.To examine data sharing and security concerns among mHealth app users—Analyzed user concerns about data privacy and security, and the factors influencing their willingness to share data with healthcare providers, focusing on how trust in the app and its security features impacted user engagement.


## 2. Material and Methods


**Study Design:** This study was designed as an observational research project to evaluate the impact of mHealth applications on lifestyle changes in individuals managing chronic health conditions, including diabetes, obesity, and hypertension. The goal was to examine the relationship between app usage and lifestyle modifications, with a focus on motivation, health awareness, and user satisfaction.**Participants:** A total of 147 individuals participated in the study. All participants in the study were of Caucasian race and Romanian ethnicity. The study was conducted exclusively in Romania. The participants’ primary language was Romanian, and all interactions with the mobile health (mHealth) application were conducted in Romanian. We recruited participants through an open invitation posted within the mHealth app platform and its affiliated social media communities between June 2024 and July 2024.


The inclusion criteria were as follows:✓Age over 18 years;✓Diagnosis of at least one of the following conditions: diabetes, obesity, or hypertension;✓Active use of the mHealth apps;✓Willingness to provide informed consent.

Sampling was conducted through a **non-probabilistic convenience strategy**, with no randomization procedures applied. Potential sources of bias include selection bias (attracting more engaged users) and self-report bias inherent to questionnaire-based assessments.

A formal **sample size calculation was not performed prior** to recruitment. Instead, all eligible and willing users were included. Post hoc power analysis indicated that the achieved sample size (*N* = 147) provided >80% power to detect moderate effect sizes (Cohen’s d = 0.5) at a significance threshold of α = 0.05.
**Exclusion Criteria:** Participants who were not using mHealth applications for health management were excluded from the study. This ensured that the sample consisted only of individuals who had direct experience with using mHealth apps.**Data Collection:** For this study, participants utilized a mobile health (mHealth) application developed by a Romanian technology company. The application was designed to support individuals managing chronic health conditions by allowing them to input and monitor a wide range of personal health data. Users could record information related to body measurements, lifestyle habits, sleeping patterns, dietary practices, exercise levels, weight management goals, blood glucose levels, and HbA1c values. The application provided tools for daily data entry, enabling users to track progress over time and identify trends in their health behaviors. Additionally, it offered personalized support features, including exercise suggestions, meal planning guidance, and healthy recipe ideas. The platform allowed for flexible and comprehensive data entry, accommodating as much information as each user wished to provide, thus promoting a personalized approach to chronic disease management. Data were collected through a structured questionnaire specifically developed for this study. The questionnaire included both closed and open-ended questions to gather information on participants’ demographics, health conditions, mHealth app usage patterns, and the perceived impact of the apps on their health and lifestyle. It was developed based on prior literature related to mobile health technology adoption and lifestyle change, and it was reviewed for face validity by two public health experts. The survey was administered online, allowing for broad accessibility, and was completed voluntarily by participants.Questionnaire Components

The questionnaire was divided into the following sections:**Demographic Information**: Age, gender, living environment, and health conditions (diabetes, obesity, hypertension).**mHealth App Usage**: Frequency of app use, primary reasons for using the app, goal-setting behavior, and app features (e.g., community support, tracking, wearable integration).**Impact on Lifestyle**: Changes in health awareness, self-monitoring, and lifestyle modifications (e.g., diet, exercise).**User Feedback**: Overall satisfaction, suggestions for app improvements, and the perceived benefits of app features (e.g., chat support, personalized recommendations).**Ethical Considerations:** The study followed ethical guidelines and received approval from the relevant institutional review board. All participants provided informed consent before taking part in the survey. The confidentiality of participant responses was maintained, and data were anonymized during analysis to protect privacy.

## 3. Results

### 3.1. Demographics


**Age**: Participants were spread across seven age groups, with the largest group being between 35–44 years (29.1%), followed by those in the 25–34 years group (18.8%). The smallest groups were the 55–64 years (10.1%) and 65+ years (10.8%) age ranges.**Gender Distribution**: Female participants made up the majority of the sample (73.5%), while male participants represented 26.5%.**Area of Residence**: Most participants lived in urban areas (82.2%), with a smaller proportion coming from rural areas (17.8%).**Education Level**: A significant proportion of participants had a college education (64.8%), while 21.3% had completed high school. Around 14.2% of participants did not disclose their educational background.**Employment Status**: The majority of participants were employed (79.9%), with retired individuals accounting for 12.2%. Smaller numbers of participants were students (6.1%) or unemployed (2.0%) (Table 1).


### 3.2. BMI Categories

The largest group of participants fell within the normal weight category, accounting for 42.9%. Overweight individuals represented 37.2% of the sample. Obesity Class 1 was reported by 11.5%, while Obesity Classes 2 and 3 were less prevalent, comprising 0.7% and 1.4%, respectively. A small portion of participants (6.1%) were classified as underweight.

### 3.3. Medical History

The majority of participants had Diabetes Mellitus (Types 1 and 2), accounting for 57.1% of the sample. Arterial hypertension and dyslipidemia were less prevalent, with 12.9% and 14.3% of participants, respectively, reporting these conditions. Obesity was reported by 15.7% of the participants.

### 3.4. mHealth Apps Usage and Perception


●**Frequency of Use**: The majority of participants (40%, or 66) reported using the mHealth app daily, suggesting it had become a key part of their routine. A smaller group (24.39%, or 40) used the app weekly, while 14.63% (24) used it occasionally, and 10.98% (17) used it rarely. These findings indicate that the app was frequently used by a core group, though there was noticeable variability in usage frequency across participants.-**User Experience and Accessibility**: When asked about the ease of accessing and navigating the app, the majority of users found it user-friendly, with 71.94% (115) selecting “easy.” A small proportion, 4.35% (7), found the app difficult to use, while 15.53% (25) were neutral. These results suggest that the app’s design was largely intuitive and easy to navigate for most participants. Regarding overall satisfaction with the app’s performance, 81.39% (130) of users reported being satisfied, while only 4.35% (7) expressed dissatisfaction. This high satisfaction rate indicates that the app generally met or exceeded users’ expectations for functionality.-**App Features**: The most valued features of the app were reminders, selected by 63.29% (100) of participants, and tracking tools, chosen by 29.63% (47). These features emphasized the app’s role in helping users stay on track with their health goals. Regarding the app’s design, 60.38% (96) of users rated it as “good,” while 15.38% (24) were neutral, and only 1.27% (2) rated it as poor. These findings indicate a strong overall appreciation for the app’s design and aesthetic quality.-**User Feedback**: Participants shared feedback on areas of the app they found problematic. Of these, 64.55% (102) cited excessive ads as a major issue. Additionally, 15.82% (25) raised concerns about functionality, and 12.69% (20) requested additional features. While the app excelled in many areas, addressing the concerns related to ads and expanding its features could enhance the overall user experience.-**Health Impact and Effectiveness**: Regarding the app’s impact on health, 81.39% (130) of users reported a positive influence, suggesting the app was generally effective in promoting better health management. To gain a deeper understanding of its effects, we further analyzed health improvements according to participants’ primary chronic conditions: diabetes, obesity, dyslipidemia, and hypertension (Table 2).


Analysis of health improvements based on participants’ primary chronic condition revealed consistent and encouraging trends. Among participants with diabetes (*N* = 84), 71 individuals (85%) reported improvements in glucose levels, and 55 individuals (65%) reported improvements in HbA1c levels. Among participants with obesity (*N* = 23), 16 (70%) reported weight loss, and 16 (68%) reported better adherence to dietary and exercise recommendations. In the dyslipidemia group (*N* = 21), 13 participants (60%) reported reductions in LDL cholesterol levels, and 12 (58%) reported improvements in triglyceride levels. In the hypertension group (*N* = 19), 11 participants (60%) reported improvements in LDL cholesterol levels, and 11 (58%) reported improvements in triglyceride levels (Figure 1).

Analysis of health outcomes by primary chronic condition revealed several statistically significant findings. Among participants with diabetes (*N* = 84), 85% reported improvements in glucose levels (*p* < 0.001) and 65% reported improvements in HbA1c levels (*p* = 0.0030), both indicating strong app-related benefits. Participants with obesity (*N* = 23) reported weight loss in 70% of cases (*p* = 0.0466) and improved adherence to dietary and exercise recommendations in 68% of cases (*p* = 0.0466), reflecting a positive lifestyle impact. In contrast, participants with dyslipidemia (*N* = 21) and hypertension (*N* = 19) showed moderate improvements in LDL cholesterol and triglyceride levels; however, these changes did not reach statistical significance (*p* > 0.05). Overall, the mHealth app demonstrated statistically significant effectiveness in managing glucose control and promoting weight loss and healthier habits among users with diabetes and obesity. These disease-specific findings suggest that the mHealth application had a significant and targeted positive impact, supporting the management of diverse chronic conditions. Although formal statistical testing was not performed, the high percentages of reported improvements across conditions highlight the app’s potential effectiveness. Future studies employing inferential statistics could further validate these preliminary observations.
-**Usefulness and Value**: Regarding overall usefulness, 43.28% (68) of users found the app helpful for managing their health, while 19.08% (30) were neutral, and 31.21% (49) did not find it useful. These findings suggest that while the app proved beneficial for many users, it may not have fully met everyone’s needs. Concerning premium features, 66.88% (105) of users believed they were worth the cost, whereas 26.58% (42) disagreed. This indicates that premium features were generally well-received by those who opted to purchase them.-**Data Sharing and Security**: Data security was a concern for 60.38% (95) of users, many of whom were reluctant to share personal information through the app. However, the same percentage ultimately shared their data with their doctors, reflecting the app’s integration into their healthcare routines. Furthermore, 60% of users reported that their doctors had approved or recommended the app, underscoring its credibility and promoting wider adoption.

## 4. Discussion

This study provides valuable insights into how users engaged with a mobile health (mHealth) app, focusing on usage frequency, user experience, health outcomes, and data sharing. It also explored how user demographics influenced app engagement. By comparing these findings with existing research, we gained a clearer understanding of broader trends in mHealth adoption, particularly how user characteristics affect health outcomes.

The demographic characteristics of the participants aligned with trends found in other mHealth studies. The majority of participants (55.81%) were aged 25–40, consistent with Gupta et al. (2019) [13], who identified this age group as the primary adopters of mHealth apps. Younger, tech-savvy adults often embrace digital tools for managing their health. However, 24.39% of participants were aged 41–55, showing significant adoption among middle-aged adults. This finding contrasts with studies such as Hamine et al. (2015) [14], where mHealth interventions mostly targeted younger demographics. This shift suggests that interest in health technology is spreading across all age groups. In line with previous research, most participants were employed (72.96%) and had at least some college education (70.71%), supporting the idea that people with higher education and stable employment are more likely to engage with health technologies [15].

The gender distribution in this study was similar to other mHealth studies, with 62.35% of participants identifying as female. Several studies have also found that women are more likely to use mobile health apps, particularly those related to wellness, fitness, and chronic disease management [16]. This gender imbalance is common and highlights an opportunity for more targeted interventions to engage male users. A recent review by Hartley et al. (2021) also noted this gap and recommended strategies to address it [17].

The study found that 40% of participants used the app daily, while 24.39% used it weekly, and 14.63% used it occasionally. This high frequency of use aligns with research indicating that frequent engagement with health apps is associated with better health outcomes, especially for chronic conditions like diabetes [18]. While there was a core group of highly engaged users, others used the app less consistently. This variability suggests that personalized strategies could help maintain engagement, especially for users who use the app intermittently. Tailoring the user experience, as suggested by Nundy et al. (2020), might improve long-term retention [19].

Most participants (71.94%) found the app easy to navigate, with 81.39% expressing satisfaction with its performance. These results align with studies emphasizing that usability and ease of navigation are key drivers of mHealth adoption. Similarly, user satisfaction is strongly linked to app usability, which is essential for long-term engagement. The small percentage (4.35%) of users who struggled with navigation indicates that while the app was generally user-friendly, there is room for improvement, particularly for users who are less familiar with technology [20]. Baig et al. (2019) suggested that tutorials or personalized onboarding processes could help address these concerns [21].

In terms of features, reminders were the most popular (63.29%), followed by tracking features (29.63%). This finding aligns with Patel’s research, which found that goal-setting and tracking functions are some of the most valued features in health apps [22]. These features are shown to effectively motivate users to stay engaged with their health management. Regarding app design, 60.38% of users rated it as “good,” consistent with studies that highlight the importance of visually appealing and intuitive design for user satisfaction and engagement. However, 64.55% of users expressed dissatisfaction with excessive advertisements, a common issue in health apps. González et al. (2020) found that intrusive ads negatively affected user experience and led to disengagement, reinforcing the need to minimize ads and prioritize a smoother user interface [23].

The positive health impact of the app was evident, with improvements in key health metrics such as glucose levels (73.42%), weight (62.02%), and adherence to dietary recommendations (71.31%). These results align with studies showing that mHealth apps are effective in managing chronic conditions like diabetes. Some studies have demonstrated that diabetes management apps improve blood glucose levels and weight [24], while others have found that mHealth interventions help users adhere to healthy lifestyle practices, especially when they include goal-setting and tracking features [25]. These broad health improvements suggest that the app was effective in promoting healthier behaviors. However, 31.21% of participants did not find the app useful, which is consistent with Schoeppe et al. (2016), who found that not all users benefit equally from mHealth interventions [26]. This variability likely reflects differences in user motivation, baseline health status, or how well the app’s features aligned with individual health goals.

Data security concerns were raised by 60.38% of participants, consistent with research identifying privacy concerns as a major barrier to mHealth adoption [27]. Despite these concerns, 60.38% of participants shared their app data with their doctors, reflecting a growing trend of integrating mHealth data into healthcare management, as noted in other studies [28]. Trust in the app’s security and approval from healthcare providers were key factors in encouraging data sharing. When users perceived their data as secure and their doctors recommended the app, they were more likely to share health information. This suggests that the app’s credibility and secure data protocols are essential for fostering trust and encouraging user engagement [29].

While the findings offer important insights, certain limitations must be acknowledged. First, the sample was recruited through convenience sampling and self-selection, which may limit the generalizability of the findings to the broader population of mHealth app users. Second, the reliance on self-reported data introduces the possibility of recall bias and social desirability bias. Third, the absence of a control group prevents causal inferences regarding the app’s impact on health outcomes. Fourth, while disease-specific analyses were conducted, adjustments for multiple comparisons were not applied, which may increase the risk of type I errors. Finally, the study did not perform a priori sample size calculation; however, post hoc analyses indicated sufficient power for detecting moderate effects.

## 5. Conclusions

Our study demonstrated that mHealth apps have a positive impact on health management, particularly in enhancing self-awareness, goal-setting, and motivation among users. Participants with chronic conditions, such as diabetes, obesity, and hypertension, reported significant improvements in key health metrics. For instance, 85% of diabetes patients noted improvements in glucose levels, and 70% of obesity patients reported weight loss. These results highlight the app’s potential in managing chronic health conditions and promoting healthier lifestyles.

However, while user satisfaction was high, several issues were identified that could limit the app’s effectiveness. Concerns regarding advertisements, data security, and varied user experiences suggest areas for improvement. Addressing these concerns could further enhance the app’s usability and support sustained engagement, potentially leading to more consistent usage and greater long-term health benefits.

Additionally, the demographic breakdown revealed that younger, educated, and employed individuals were more likely to engage with the app. To broaden the app’s reach, particularly among older age groups and male users, targeted outreach efforts may be needed. Increasing adoption across different segments could amplify the app’s impact on public health.

In conclusion, our study supports the use of mHealth apps as valuable tools for managing chronic health conditions. However, incorporating user feedback and expanding the app’s features could increase its effectiveness and appeal, fostering greater user engagement and long-term health improvements.

## Figures and Tables

**Figure 1 nutrients-17-01506-f001:**
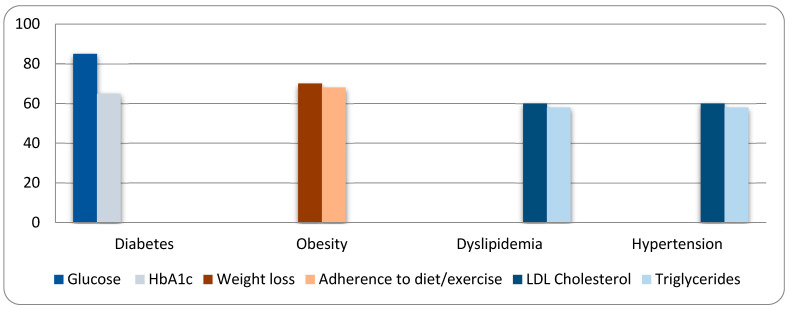
Data for health metrics improvement.

**Table 1 nutrients-17-01506-t001:** Demographic characteristics of study participants.

Demographic Category	Details	Percentage (%)
Age Groups	18–24 years	12.3%
	25–34 years	18.9%
	35–44 years	29.1%
	45–54 years	18.8%
	55–64 years	10.1%
	65+ years	10.8%
Gender Distribution	Female	73.5%
	Male	26.5%
Area of Residence	Urban	82.2%
	Rural	17.8%
Education Level	College Education	64.8%
	High School	21.3%
	No Education/Undisclosed	14.2%
Employment Status	Employed	79.9%
	Retired	12.2%
	Student	6.1%
	Unemployed	2.0%

**Table 2 nutrients-17-01506-t002:** Health improvements reported by participants, according to primary chronic condition.

Condition	Improvement Indicator	Percentage (%)	*N* (Number of Participants)
Diabetes (*N* = 84)	Improved glucose levels	85%	71
	Improved HbA1c levels	65%	55
Obesity (*N* = 23)	Weight loss	70%	16
	Better adherence to diet/exercise recommendations	68%	16
Dyslipidemia (*N* = 21)	Reduction in LDL cholesterol	60%	13
	Improvement in triglyceride levels	58%	12
Hypertension (*N* = 19)	Reduction in LDL cholesterol	60%	11
	Improvement in triglyceride levels	58%	11

## Data Availability

The datasets presented in this article are part of a larger, ongoing research project and are not publicly available at this time. Data may be made available upon reasonable request to the corresponding author and with permission from the principal investigators of the overarching study.

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
