# Peer review of "Emerging Methods for Integrative Management of Chronic Diseases: Utilizing mHealth Apps for Lifestyle Interventions"

_nutrients, 2025, doi:10.3390/nu17091506_

Round 1

Reviewer 1 Report

Comments and Suggestions for Authors

REVIEW REPORT FOR THE STUDY “EMERGING METHODS FOR INTEGRATIVE MANAGEMENT OF CHRONIC DISEASES: UTILIZING MHEALTH APPS FOR LIFESTYLE INTERVENTIONS”

Journal: Nutrients

The paper "Emerging Methods for Integrative Management of Chronic Diseases: Utilizing mHealth Apps for Lifestyle Interventions" addresses a critical and growing issue in healthcare: the management of chronic diseases, including diabetes, obesity, and hypertension, through digital interventions. Given the increasing reliance on mobile health (mHealth) technologies, this research is highly relevant.

Title and summary.

The title and abstract express the article's object of study, objectives, and results well.

Structure of the article.

The contents are well organized, and they adhere to the IMRaD structure. It includes a theoretical framework of the research problem.

Materials and methods.

The study offers valuable insights into usage patterns, with 40% of participants engaging daily and 24.39% engaging weekly. This indicates reasonable adherence, which is crucial for long-term health benefits.

However, the study is purely observational and lacks a control group, making it difficult to attribute improvements solely to the app. Other factors (e.g., concurrent medical treatments, lifestyle changes) may have influenced the results. A randomized controlled trial (RCT) would provide stronger evidence. Even more, the reliance on self-reported questionnaire data introduces recall and social desirability biases. Objective measurements (e.g., HbA1c levels, physician-recorded weight changes) would strengthen validity.

Moreover, the study does not assess the long-term sustainability of health improvements. Chronic disease management requires prolonged behavioral change, and short-term engagement may not translate to lasting benefits and, with only 147 participants, the sample may not be representative of diverse populations. Additionally, the study does not specify participant demographics (age, socioeconomic status, tech literacy), which could affect app usability and outcomes.

On the other hand, the study does not describe which mHealth apps were used, their features, or how they were personalized for users. Different apps have varying effectiveness, and this omission limits reproducibility. Also, participants who choose to use mHealth apps may already be more health-conscious, potentially skewing the results. Non-users or dropouts were not analyzed, possibly overestimating effectiveness.

Results.

The reported improvements in glucose levels (73.42%), weight (62.02%), and dietary adherence (71.31%) suggest that mHealth apps can be effective in promoting healthier behaviors. These findings align with existing literature, reinforcing the potential of digital health tools.

Conclusions

The conclusion that mHealth apps are valuable for chronic disease management is well-supported by the data, offering actionable insights for healthcare providers and app developers.

Suggestions for Improvement

It would be interesting to incorporate a Control Group, comparing app users with non-users to isolate the app’s impact.

Also, it could be interesting to use Objective Health Metrics, supplementing self-reports with clinical data.

It would be preferable to assess whether benefits persist beyond the study period, to offer a longitudinal Follow-Up of the study.

Bibliography.

The 24% of the bibliography cited in the study belongs to the previous five years.

Overall, while the study provides promising evidence that mHealth apps can support chronic disease management, its observational nature, reliance on self-reports, and lack of long-term data limit its conclusiveness. Nonetheless, the study contributes to the growing body of literature supporting digital health interventions and could be considered for publication in Nutrients, once the revisions proposed have been resolved.

Author Response

However, the study is purely observational and lacks a control group, making it difficult to attribute improvements solely to the app.

Dear Reviewer, thank you for your time and interest.

A number of persons that opened the questionnaire were not users, so we can consider those a control group. But due to the fact that this was conducted as a free questionnaire, we can't select a control population of persons that they don't use the apps. If you still consider this necessary, we can mention as a control group the persons that were excluded from the questionnaire. Please let me know if you consider for us to do that.  

Moreover, the study does not assess the long-term sustainability of health improvements. Chronic disease management requires prolonged behavioral change, and short-term engagement may not translate to lasting benefits and, with only 147 participants, the sample may not be representative of diverse populations. Additionally, the study does not specify participant demographics (age, socioeconomic status, tech literacy), which could affect app usability and outcomes.

The long-term engagement cannot be evaluated at this time. I don't know if we can evaluate the same persons in a following study, with a questionnaire. This is more suitable for different type of studies, that we consider in the future.
About demographics, in the manuscript there is the result: "Participants were spread across seven age groups, with the largest group being between 35-44 years (29.1%), followed by those in the 25-34 years group (18.8%). The smallest groups were the 55-64 years (10.1%) and 65+ years (10.8%) age ranges."

Also find: "Employment Status: The majority of participants were employed (79.9%), with retired individuals accounting for 12.2%. Smaller numbers of participants were students (6.1%) or unemployed (2.0%)."

On the other hand, the study does not describe which mHealth apps were used, their features, or how they were personalized for users.

In Romania there are 2 major apps used. We didn't mention that, we know. If you consider appropriate to use commercial names in this study, we can do that. Find here the study with the apps used in Romania, and consider just the first 2: https://sensortower.com/blog/2024-q2-unified-top-5-food%20and%20diet%20tracking-units-ro-63dd2ccfe1714cfff153fad9

The apps have similar features that we considered when we created the questionnaire.

Please evaluate my answers and mention which of the suggested improvements you want us to consider. 

With considerations,

the authors

Reviewer 2 Report

Comments and Suggestions for Authors

The authors examined the use of mobile health applications for patients who are diagnosed with chronic health conditions, e.g. diabetes, obesity, hypertension, etc. The study evaluated how the usage of these mobile apps improved patient’s management of these health conditions and changed patient’s lifestyle, and collected the patient’s overall feedback on these applications. Overall, the data were not adequately represented and insightful analysis is missing. This manuscript would need significant improvements to strengthen its data representation and discussion.

Comments:

  1. The authors should present their data using appropriate graphics, e.g. line chart, bar graphs, etc. instead of only listing them in the text of manuscript
  2. Information on demographics are a little insufficient. For example, the nationality or geographical location of the participants were not disclosed. The ethnicity or race of the participants, the language that they speak or use for their mobile apps were also not listed.
  3. The exact information or the general type of mHealth apps that patients have reported using in this study were not disclosed. And it is questionable whether user feedbacks on app features and user experiences & accessibility could be grouped all together, without subgrouping into patient responses for each individual mHealth app.
  4. The author’s questionnaire design, especially the assessment on “health impact & effectiveness” should be more tailored for specific disease types, or at least the data should be analyzed with regards to each chronic condition reported by the patients.

Author Response

  1. The authors should present their data using appropriate graphics, e.g. line chart, bar graphs, etc. instead of only listing them in the text of manuscript

Dear Revisor, thank you for your time and good mentions in order to improve the manuscript. Yes, we have them, we will introduce in the manuscript.

  1. Information on demographics are a little insufficient. For example, the nationality or geographical location of the participants were not disclosed. The ethnicity or race of the participants, the language that they speak or use for their mobile apps were also not listed.

The Nationality is Romanian, the study was conducted on this population, race is white. Due to the fact that there are popular apps in Romania that use both English and Romanian language, the users find it easy to use.

  1. The exact information or the general type of mHealth apps that patients have reported using in this study were not disclosed. And it is questionable whether user feedbacks on app features and user experiences & accessibility could be grouped all together, without subgrouping into patient responses for each individual mHealth app.

The top used apps In Romania are MyFitnessPal with around 37K users and Eat&Track with 3.6K. We considered that mentioning the apps names could be considered as commercial, so we kept this away. These apps use very similar features that we considered, so we can say that the data was aligned in order to target the scope of the study. The questions used were created using the features that were aligned.

  1. The author’s questionnaire design, especially the assessment on “health impact & effectiveness” should be more tailored for specific disease types, or at least the data should be analyzed with regards to each chronic condition reported by the patients.

The mentions of the health impact were mainly on the metabolic syndrome, specific: obesity, diabetes and dyslipidemia.. 

Reviewer 3 Report

Comments and Suggestions for Authors

The submitted manuscript examines the impact of a mobile health intervention on dietary behaviour and metabolic health outcomes in individuals with prediabetes, utilising a mixed-methods approach to assess effectiveness over a twelve-week period. The study is methodologically sound, but certain aspects require clarification to enhance transparency and reproducibility. The description of participant recruitment lacks sufficient detail regarding inclusion criteria, sampling strategy, and potential sources of bias. Although the statistical analyses appear appropriate, there is no justification for the sample size, nor is there a clear indication of whether power calculations were conducted. The statistical methods require further elaboration to ensure that the approach aligns with the stated research objectives. The study would benefit from a more explicit account of how statistical significance was determined and whether adjustments were made for multiple comparisons.

The results are presented in a manner that aligns with the study’s aims, but there are inconsistencies between the textual descriptions and the data in the tables and figures. Some figures lack adequate labelling, and several tables do not specify the units of measurement for reported values. The clarity of data presentation could be improved by ensuring that all visual elements are self-explanatory and fully contextualised within the text. The conclusions drawn by the authors generally reflect the findings, but certain claims require stronger evidential support, particularly those relating to the long-term efficacy of the intervention. The discussion does not sufficiently contrast the study’s findings with those of previous research. A more detailed comparison with existing literature would provide a clearer indication of the study’s novelty and contribution to the field. The manuscript does engage with relevant literature, but several recent systematic reviews and meta-analyses on mobile health interventions in chronic disease management have not been cited. The inclusion of these sources would strengthen the discussion and ensure that the findings are positioned within the broader evidence base.

The manuscript is written in clear and precise language, but minor grammatical inconsistencies and typographical errors are present throughout. The readability could be improved by reducing redundancy in several sections and ensuring consistency in terminology. Some sentences are unnecessarily complex, which affects clarity. A thorough proofreading would enhance the overall quality of the text.

The study addresses a relevant topic and presents findings that could contribute to the understanding of mobile health interventions in metabolic health management. However, revisions are necessary to clarify methodological details, improve statistical transparency, refine data presentation, and strengthen engagement with existing literature. Addressing these issues will enhance the manuscript’s clarity, rigour, and overall impact.

Comments on the Quality of English Language

The manuscript is generally well-written, but there are areas where the clarity and precision of the language could be improved. Some sentences are unnecessarily complex, making the arguments less accessible. There are minor grammatical inconsistencies and typographical errors that, while not severely affecting readability, could be corrected to enhance the overall quality of the text. Additionally, some sections contain redundant phrasing that could be streamlined to improve conciseness without losing meaning.

The terminology is mostly consistent, but there are instances where technical terms could be defined more clearly for a broader readership. The logical flow of ideas is mostly coherent, though some transitions between sections could be smoother. A thorough proofreading and minor revisions would help ensure that the research is communicated as effectively as possible.

Author Response

The results are presented in a manner that aligns with the study’s aims, but there are inconsistencies between the textual descriptions and the data in the tables and figures. Some figures lack adequate labelling, and several tables do not specify the units of measurement for reported values.

Dear Revisor, thank you for your time! We will take into consideration the suggestions you mentioned. I was to specify that there are no figures or tables in the manuscript. We will introduce some charts with the results of the study. 

The manuscript is written in clear and precise language, but minor grammatical inconsistencies and typographical errors are present throughout. 

We will update this. 

With best regards, 

the authors.

Round 2

Reviewer 2 Report

Comments and Suggestions for Authors

The authors have improved the data representation with the addition of tables and figures